# Qualitative and Quantitative Detection of Multiple Sexually Transmitted Infection Pathogens Reveals Distinct Associations with Cervicitis and Vaginitis

Wen-Tyng Kang,[a,b,c] Haibo Xu,[a,b,c] Yiqun Liao,[a,b,c] Qiwei Guo,[d,e] Qiuying Huang,[a,b,c] Ye Xu,[a,b,c] Qingge Li[a,b,c]

[a]Engineering Research Centre of Molecular Diagnostics, Ministry of Education, State Key Laboratory of Cellular Stress Biology, School of Life Sciences, Xiamen University, Xiamen, Fujian, China
[b]State Key Laboratory of Molecular Vaccinology and Molecular Diagnostics, School of Public Health, Xiamen University, Xiamen, Fujian, China
[c]Engineering Research Centre of Personalized Molecular Diagnostics of Xiamen, Xiamen, Fujian, China
[d]Xiamen Maternal and Child Health Hospital, Xiamen, Fujian, China
[e]United Diagnostic and Research Center for Clinical Genetics, School of Public Health of Xiamen University, Xiamen, Fujian, China

**ABSTRACT** Many diverse pathogens have been discovered from reproductive-tract infections, but the relationship between the presence and abundance of particular pathogen species and disease manifestations is poorly defined. The present work examined the association of multiple common pathogens causing sexually transmitted infections (STIs) with cervicitis and vaginitis. The presence and abundance of 15 STI pathogens and the genotypes of human papillomavirus were determined in a cohort of 944 women that included 159 cervicitis patients, 207 vaginitis patients, and 578 healthy controls. Logistic regression and random forest models were constructed and validated in a separate cohort of 420 women comprising 52 cervicitis patients, 109 vaginitis patients, and 259 healthy controls. The frequency of individual STI pathogen species varied among the symptomatic patients and healthy controls. Abundance determination was necessary for most pathogens that were associated with the studied diseases. STI pathogens were more commonly associated with cervicitis than with vaginitis. Pathogen identification- and quantification-based diagnosis was observed for cervicitis with high sensitivity and specificity, but for vaginitis, the assay results would need to be combined with results of other diagnostic tests to firmly establish the pathogen-disease correlation. Integrated qualitative and quantitative detection of a selected panel of common STI pathogens can reveal their association with cervicitis and vaginitis. STI pathogen identification and quantification can be used to diagnose cervicitis and also help improve correct diagnosis of vaginitis.

**IMPORTANCE** Scarce information exists with regard to whether STI pathogens can be defined as valid microbiological predictive markers for the diagnosis of cervicitis and vaginitis. We therefore conducted this study to assess the presence and abundance of a wide range of STI pathogens among patients having these two diseases and healthy controls as well. High sensitivity and specificity were observed for cervicitis by pathogen identification- and quantification-based diagnosis. In contrast, the assay results obtained for vaginitis would need to be combined with test results obtained by other diagnostic methods to decisively establish the pathogen-disease correlation. Simultaneous qualitative and quantitative detection of a selected panel of common STI pathogens and further coupling with machine learning models is worthwhile for establishing pathogen-based diagnosis of gynecological inflammations, which could be of great value in guiding the rational use of antimicrobials to control the spread of STIs.

**KEYWORDS** sexually transmitted infections, cervicitis, logistic regression, qualitative and quantitative detection, random forest, vaginitis

Address correspondence to Qingge Li, qgli@xmu.edu.cn.
The authors declare no conflict of interest.

Sexually transmitted infections (STIs) are commonly associated with gynecological inflammation, particularly in reproductive-age women (1). In 2016, at least 376 million cases worldwide were reported for the four curable STIs (chlamydiosis, gonorrhea, trichomoniasis, and syphilis), which account for more than one million infections per day (2). Numbers of these STIs have been on a steady rise, reaching 689 million in 2019 (3). Concurrent with the increase in these classic STIs are emerging outbreaks of "non-classical" sexually transmissible pathogens (4). STIs increase the rates of infertility, chronic pelvic pain, ectopic pregnancy, miscarriage, fetal death, congenital and neonatal infections, and HIV acquisition (5). Treatment and intervention of these infections rely on diagnosis at the pathogen level. Unfortunately, pathogen-based diagnosis of STIs has long been hindered by factors such as nonspecific clinical symptoms, wide diversity of microbe species, coinfection, etc. Thus, syndromic management is often recommended on the basis of observed symptoms, e.g., vaginal discharge, urethral discharge, genital ulcers, abdominal pain, etc. (6). Syndromic management in the absence of laboratory tests can result in misdiagnosis of STIs among women at an alarming rate, as the algorithms fail to diagnose nearly 80% of chlamydial, gonococcal, and trichomoniasis infections (7–9).

Development of pathogen-based diagnosis relies on the etiological association of syndrome with pathogen. Establishing such an association is challenging for STIs because many previous conclusions regarding their infectious causes were derived from studies that used a variety of methods for pathogen detection, ranging from microscopy to antigen detection to PCR (10). Each method has its own sensitivity and specificity, which could substantially affect conclusions. Moreover, many STIs present asymptomatically (11). This is especially true for women, who are at the same time most at risk for reproductive-health complications. Few studies have attempted to explore the cutoff values based on quantitative pathogen detection to differentiate patients and healthy controls. Even in the era of metagenomic next-generation sequencing, studies on the association of STI pathogens with syndrome are rare (12). Pathogen-specific syndromic testing for STIs is largely unavailable, despite the wide use of pathogen-based syndromic testing assays for other infectious diseases.

In the present study, we explored the association between common STI pathogens and two major syndromes in gynecological inflammation: cervicitis and vaginitis. Cervicitis is a condition presenting as inflammation of the uterine cervix caused by infection and represents approximately 40% of women seen in consultations for STIs (13–15). Vaginitis occurs when there is an overabundance of particular microbes in the vagina, resulting in changes in the balance of the microbiota in this microenvironment (16). Vaginitis is the most prevalent cause of abnormal vaginal discharge in women of childbearing age. We used a multiplex PCR assay to screen for 15 common STI pathogens and then individual singleplex quantitative real-time PCR (qPCR) assays for more detailed quantification. The infection rates and relative abundances of the pathogens were compared in a cohort that included patients having the two conditions and healthy controls. The combined qualitative and quantitative data were then used to construct predictive models using logistic regression and the random forest algorithm to further explore the association of STI pathogens with cervicitis and vaginitis. Finally, we determined the sensitivity and specificity of the models in a separate cohort.

## RESULTS

**Study overview and participants.** As illustrated in the flowchart (Fig. 1), our study consisted of three major parts. First, a 16-plex MeltArray assay was used for screening of the 15 STI pathogens, including *Chlamydia trachomatis*, *Neisseria gonorrhoeae*, *Mycoplasma genitalium*, *Mycoplasma hominis*, *Ureaplasma urealyticum*, *Ureaplasma parvum*, *Treponema pallidum*, *Haemophilus ducreyi*, *Klebsiella granulomatis*, *Trichomonas vaginalis*, human cytomegalovirus (HCMV), herpes simplex virus 1 (HSV-1), herpes simplex virus type 2 (HSV-2), varicella-zoster virus (VZV), and human herpesvirus 8 (HHV-8). A human papillomavirus (HPV) genotyping assay that cover 37 genotypes was used for screening for HPV. Individual singleplex qPCR assays were used to determine the

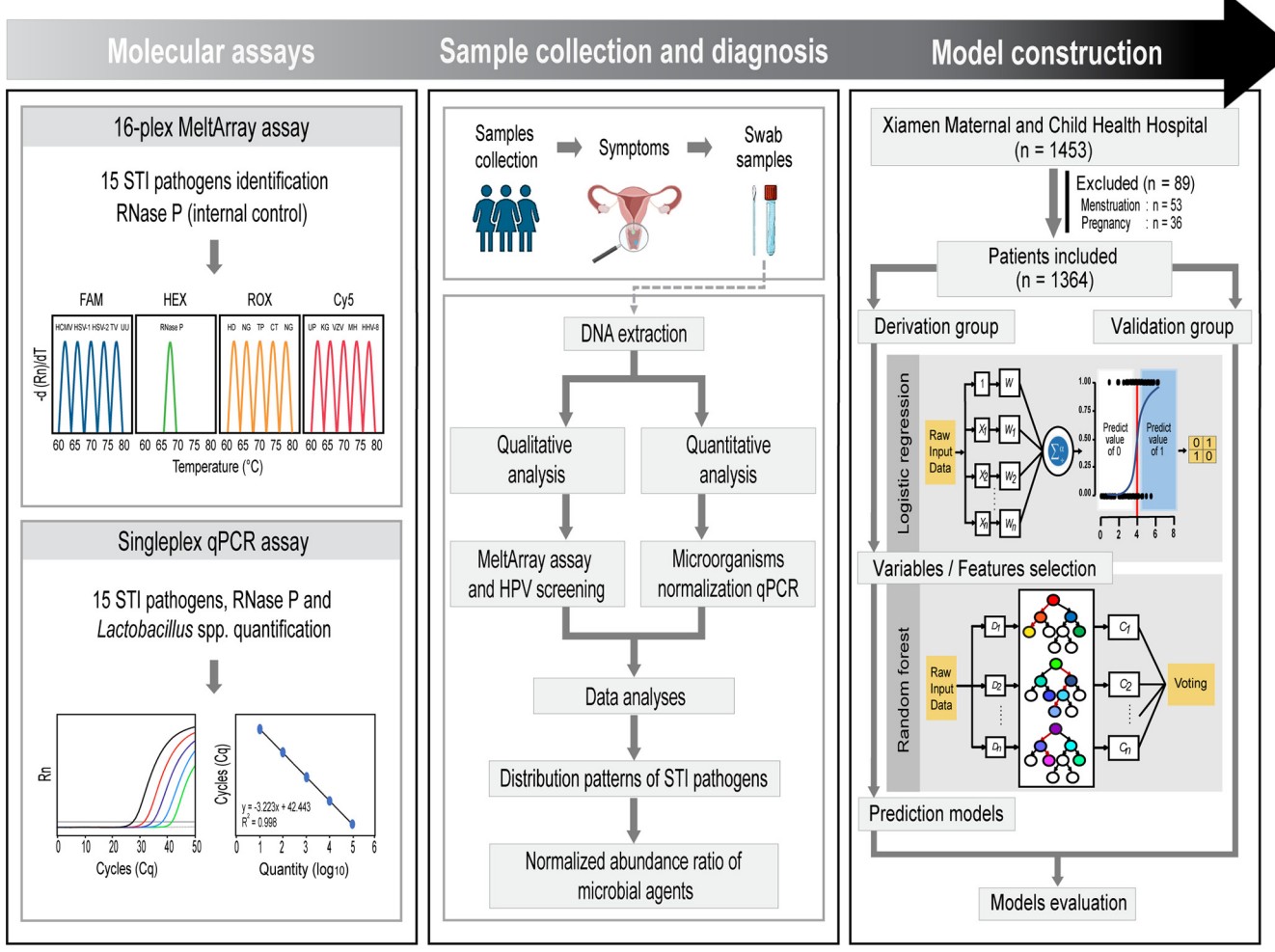

**FIG 1** Flowchart for three study phases: molecular assays, sample collection and diagnosis, and model construction. Phase I describes the 16-plex MeltArray screening assay and singleplex quantitative real-time PCR assays. Phase II shows sample collection and diagnosis of cervicitis and vaginitis, with a summary of laboratory procedures. Phase III demonstrates the enrollment of both the derivation and validation groups for constructing and validating the prediction models.

abundance of the 15 STI pathogens and *Lactobacillus* spp. relative to the human RNase P gene. Second, the qualitative data were used to compare the infection rate of each pathogen; the quantitative data were used to compare relative abundance of each pathogen. Third, model constructions were performed using logistic regression and the random forest algorithm using the combined qualitative and quantitative data. The models were then validated regarding the sensitivity and specificity. For the study populations, a total of 1,453 participants were initially included. Of these, 89 of them were excluded and 1,364 participants were ultimately included according to the eligibility criteria. The derivation cohort contained 944 women with a median age of 36 (interquartile range [IQR], 30 to 43) years, consisting of 578 (61.2%) healthy women, 159 (16.8%) cervicitis patients, and 207 (21.9%) vaginitis patients. The validation cohort contained 420 women with a median age of 32 (IQR, 27 to 39) years, consisting of 259 (61.7%) healthy women, 52 (12.4%) cervicitis patients, and 109 (25.9%) vaginitis patients.

**Molecular assays.** The singleplex qPCR assay for the 15 STI pathogens, *Lactobacillus* spp., and human RNase P displayed equivalent analytical performance regarding the limit of detection (LOD) ($10^0$ copy/$\mu$L) and quantification range ($R^2$ = 0.993 to 0.999) (see Table S1 in the supplemental material). These assays were used as reference methods for pathogen detection; further clinical evaluations were omitted owing to their widely recognized performances in clinical use. Concordance between the 16-plex MeltArray assay and singleplex qPCR assays was initially evaluated using a collection of

203 vaginal swab samples for 203 MeltArray reactions and 3,248 singleplex reactions. The overall concordance between the two assays was ≥97.8% (Fig. S1). As certain pathogens (e.g., *H. ducreyi*, HSV-1, and *T. vaginalis*) were rarely detected, the results from the derivation cohort containing 944 samples were used for further comparison. This work involved 944 MeltArray reactions and 15,104 singleplex reactions. The overall results yielded a concordance of 98.4% between the two assays. No significant difference was found in detecting the presence of each target (Table S2). Notably, the 16-plex MeltArray assay could generate qualitative results for the 16 targets from a sample in a single reaction within 2.5 h. In contrast, the singleplex qPCR assay required 16 times the number of assays to generate normalized abundance ratio (NAR) for each pathogen.

**Infection frequency of STI pathogens among three study groups.** The overall STI pathogen infection frequency was 78.4% (740/944) for the entire cohort. Both cervicitis (91.8%, 146/159) and vaginitis (90.8%, 188/207) patients exhibited higher infection rates than the healthy controls (70.2%, 406/578) (Fig. 2A, inset). The seven most prevalent pathogens were *U. parvum* (42.6%, 402/944), HCMV (24.2%, 228/944), HPV (15.7%, 148/944), *C. trachomatis* (15.4%, 145/944), *M. hominis* (12.7%, 120/944), HHV-8 (12.4%, 117/944), and *K. granulomatis* (10.8%, 102/944). These seven pathogens accounted for 86.1% of all pathogens identified (Fig. 2A). Pathogens that showed an association with cervicitis and vaginitis are listed in Table 1. Six pathogens, *C. trachomatis* (adjusted odds ratio [aOR], 2.78; 95% confidence interval [CI], 1.60 to 4.83), *K. granulomatis* (aOR, 2.40; 95% CI, 1.34 to 4.30), *T. pallidum* (aOR, 19.76; 95% CI, 5.19 to 75.13), *N. gonorrhoeae* (aOR, 28.11; 95% CI, 5.54 to 142.47), *H. ducreyi* (aOR, 12.27; 95% CI, 1.05 to 143.59), and HPV (aOR, 2.43, 1.47 to 4.00), showed significant associations with cervicitis. Seven pathogens, *C. trachomatis* (aOR, 5.09; 95% CI, 3.17 to 8.17), *M. hominis* (aOR, 2.03; 95% CI, 1.22 to 3.39), *K. granulomatis* (aOR, 2.36; 95% CI, 1.38 to 4.04), *T. pallidum* (aOR, 23.79; 95% CI, 6.54 to 86.48), *N. gonorrhoeae* (aOR, 7.20; 95% CI, 1.33 to 38.93), HCMV (aOR, 1.57; 95% CI, 1.05 to 2.34), and HHV-8 (aOR, 1.79; 95% CI, 1.09 to 2.95), were associated with vaginitis. Among the above pathogens, four, *C. trachomatis*, *K. granulomatis*, *T. pallidum*, and *N. gonorrhoeae*, were found in both patient populations with higher infection risk than in the healthy subjects. *H. ducreyi* and HPV had a higher risk in the cervicitis patients, while *M. hominis*, HCMV, and HHV-8 had increased risk for vaginitis.

Coinfections were found in 47.8% (451/944) of the entire study population. Cervicitis and vaginitis had an average of 2.1 and 2.3 pathogens per sample, respectively, compared to 1.7 pathogens per sample in healthy subjects. Coinfections with more than 3 pathogens were more frequently found in the patients than in the healthy subjects. Vaginitis tended to have higher coinfections of 3 and 4 pathogens than cervicitis, as seen from their individual *P* values (Fig. 2B and Fig. S2A). Overall, cervicitis (59.7%, 95/159) and vaginitis (64.7%, 134/207) patients had higher coinfection rate than the healthy subjects (38.4%, 222/578, $P < 0.0001$). No significant difference was observed between cervicitis and vaginitis ($P = 0.329$) (Fig. 2C). When results were examined with Spearman's correlation coefficients, no obvious correlation was observed among the majority of the pathogens in the three groups despite a few positive correlations (Fig. S2B). These results indicate that although coinfections with ≥3 STI pathogens were more frequently found in the patients, most of the STI pathogens contributed independently to the two conditions.

**Quantitative analysis of the STI pathogens and *Lactobacillus* spp. among three study groups.** According to the median NAR, *M. genitalium*, *U. urealyticum*, *M. hominis*, and VZV were significantly more abundant in both cervicitis and vaginitis patients than in the healthy subjects (Fig. 3A and Table S3). A lower median NAR (mNAR) of *Lactobacillus* spp. was found in both cervicitis (mNAR = 0.76) and vaginitis (mNAR = 0.75) patients than in healthy subjects (mNAR = 0.83). A higher median NAR of HCMV was present only in vaginitis patients. Hierarchical clustering analysis of STI pathogens and *Lactobacillus* spp. for the entire population showed that *Lactobacillus* spp. were dominant exclusively in the healthy subjects. Both cervicitis and vaginitis triggered a shift from a *Lactobacillus* spp.-dominant state to an STI pathogen-dominant state (Fig. 3B). Using NAR cutoff values to stratify the patients from healthy subjects revealed that most microbes with a NAR above

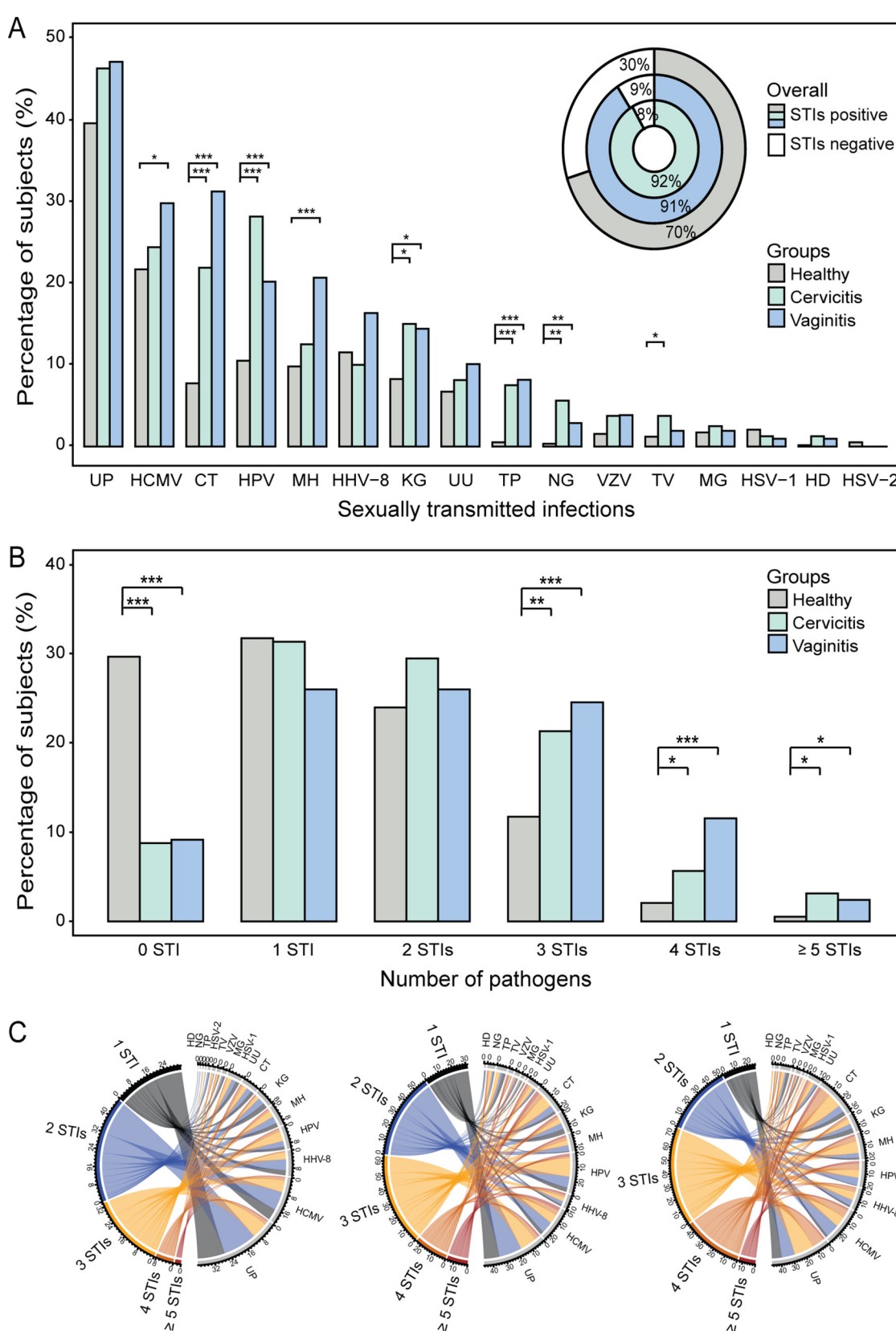

**FIG 2** Frequencies of the STI pathogens in healthy subjects and those with cervicitis and vaginitis. (A) The pie chart shows the percentages of STI-positive and STI-negative cases, and the bar chart represents the overall prevalence of STIs for all the study

the cutoff values were significantly associated with cervicitis and vaginitis (Fig. 3C and Table 2).

In univariable analyses for each microbe, except for HCMV, HHV-8, HSV-1, and HSV-2, all STI pathogens and *Lactobacillus* spp. showed significant association with cervicitis. In contrast, except for *H. ducreyi*, HSV-1, and HSV-2, all STI pathogens and *Lactobacillus* spp. were associated with vaginitis patients. In multivariable analyses for microbes with NARs above the cutoff values, *Lactobacillus* spp. (aOR, 0.36; 95% CI, 0.23 to 0.58) and five STI pathogens, including *M. genitalium* (aOR, 8.42; 95% CI, 1.53 to 46.45), *U. urealyticum* (aOR, 2.90; 95% CI, 1.05 to 8.00), *C. trachomatis* (aOR, 8.74; 95% CI, 1.29 to 59.11), *M. hominis* (aOR, 6.29; 95% CI, 1.82 to 21.72), and *U. parvum* (aOR, 2.45; 95% CI, 1.47 to 34.07), were strongly associated with cervicitis. *Lactobacillus* spp. (aOR, 0.44, 95%CI 0.29 to 0.65) and six STI pathogens, including *U. urealyticum* (aOR, 2.91; 95% CI, 1.10 to 7.70), *T. pallidum* (aOR, 43.23; 95% CI, 5.33 to 350.67), *C. trachomatis* (aOR, 4.42; 95% CI, 2.49 to 7.83), *K. granulomatis* (aOR, 4.65; 95% CI, 1.37 to 15.82), *M. hominis* (aOR, 10.40; 95% CI, 3.37 to 32.09), and HHV-8 (aOR, 2.53; 95% CI, 1.32 to 4.83), showed significant association with vaginitis (Table 3).

**Logistic regression examination of the association of STI pathogens with patients.** Multivariable analysis of the derivation group (944 samples), after adjustment for potential confounder effects, confirmed that eight STI pathogens were significantly associated with cervicitis (qualitatively for *N. gonorrhoeae*, *T. pallidum*, HPV, *C. trachomatis*, and *K. granulomatis*; quantitatively for *M. genitalium*, *M. hominis*, and *U. parvum*). Nine pathogens were associated with vaginitis (qualitatively for *N. gonorrhoeae*, *C. trachomatis*, and *K. granulomatis*; quantitatively for *T. pallidum*, *M. hominis*, *T. vaginalis*, *U. urealyticum*, HHV-8, and HCMV) (Fig. 4A and Table S4). The largest AUCs for cervicitis (0.80; 95% CI, 0.77 to 0.83) and vaginitis (0.78; 95% CI, 0.75 to 0.81) were obtained by integrating both qualitative and quantitative data sets rather than with either of them alone (Table S5). The predictive value of the validation group (420 samples) showed that the sensitivity, specificity, and AUC were 0.90, 0.71, and 0.85 (95% CI, 0.80 to 0.90), respectively, for the cervicitis patients; they were 0.60, 0.76, and 0.72 (95% CI, 0.66 to 0.78), respectively, for the vaginitis patients (Fig. 4B and C). The goodness of fit can indicate that a predictive model is a good fit to the data set if a $P$ value of $>0.05$ is obtained through the Hosmer-Lemeshow test. Accordingly, the predictive models for cervicitis with both the derivation and the validation groups had a good calibration with goodness of fit (Hosmer-Lemeshow, $\chi^2$) of 13.56 ($P = 0.060$) and 11.64 ($P = 0.113$), respectively. In contrast, the prediction models of vaginitis with the derivation group had a good calibration, with goodness of fit of 9.51 ($P = 0.218$), whereas it lacked consistency with the validation group in the calibration with goodness of fit of 14.67 ($P = 0.040$) (Fig. S3).

**Random forest analysis of the importance of STI pathogens in patients.** Feature selection was performed with the random forest algorithm through incorporation of both the qualitative and quantitative data sets. The Gini importance was used to measure feature relevance (Fig. 5A). The results showed that 12 STI pathogens were associated with cervicitis (qualitatively for HPV, *C. trachomatis*, *T. pallidum*, *N. gonorrhoeae*, *K. granulomatis*, HHV-8, and VZV; quantitatively for *U. parvum*, VZV, *M. genitalium*, *T. pallidum*, *M. hominis*, *U. urealyticum*, *N. gonorrhoeae*, *H. ducreyi*, and *C. trachomatis*). Notably, *C. trachomatis*, *T. pallidum*, *N. gonorrhoeae*, and VZV were included both qualitatively and quantitatively. Eleven pathogens were associated with vaginitis (qualitatively for *C. trachomatis*,

**FIG 2** Legend (Continued)

participants. *, $P < 0.05$; **, $P < 0.001$; ***, $P < 0.0001$. (B) Single infection and coinfections by 16 STI pathogens among cervicitis patients versus healthy subjects and vaginitis patients versus healthy subjects. (C) Chord diagrams representing the coinfection patterns of detected STIs in the healthy, cervicitis, and vaginitis groups. Types of infections among healthy and patient groups include single infection and coinfections with up to six STI pathogens. Coinfection patterns of STI pathogens in the cervicitis and vaginitis groups show a trend of diversification and complexity in comparison to the healthy groups. Abbreviations: UP, *Ureaplasma parvum*; HCMV, human cytomegalovirus; CT, *Chlamydia trachomatis*; HPV, human papillomavirus; MH, *Mycoplasma hominis*; HHV-8, human herpesvirus 8; KG, *Klebsiella granulomatis*; UU, *Ureaplasma urealyticum*; TP, *Treponema pallidum*; NG, *Neisseria gonorrhoeae*; VZV, varicella-zoster virus; TV, *Trichomonas vaginalis*; MG, *Mycoplasma genitalium*; HSV-1, herpes simplex virus 1; HD, *Haemophilus ducreyi*; HSV-2, herpes simplex virus type 2.

**TABLE 1** Unadjusted and adjusted odds ratio showing the association between STI pathogens and women with cervicitis and vaginitis

| Variable | No. (%) of: All subjects (n = 944) | No. (%) of: Healthy subjects (n = 578) | Cervicitis (n = 159) No. (%) | OR[a] (95% CI) | P[b] | aOR[c] (95% CI) | P[d] | Vaginitis (n = 207) No. (%) | OR[a] (95% CI) | P[b] | aOR[c] (95% CI) | P[d] |
|---|---|---|---|---|---|---|---|---|---|---|---|---|
| **Age (yrs)** | | | | | | | | | | | | |
| 16–25 | 59 (6.3) | 14 (2.4) | 20 (12.6) | Reference | | Reference | | 25 (12.1) | Reference | | Reference | |
| 26–35 | 397 (42.1) | 247 (42.7) | 61 (38.4) | 0.17 (0.08–0.36) | **<0.0001** | 0.22 (0.10–0.50) | **<0.0001** | 89 (43.0) | 0.20 (0.10–0.41) | **<0.0001** | 0.32 (0.15–0.72) | **0.005** |
| 36–45 | 310 (32.8) | 194 (33.6) | 54 (34.0) | 0.20 (0.09–0.41) | **<0.0001** | 0.29 (0.12–0.66) | **0.003** | 62 (30.0) | 0.18 (0.09–0.37) | **<0.0001** | 0.29 (0.13–0.66) | **0.003** |
| 46–55 | 142 (15.0) | 97 (16.8) | 21 (13.2) | 0.15 (0.07–0.35) | **<0.0001** | 0.19 (0.07–0.48) | **<0.0001** | 24 (11.6) | 0.14 (0.06–0.31) | **<0.0001** | 0.23 (0.09–0.56) | **0.001** |
| ≥56 | 36 (3.8) | 26 (4.5) | 3 (1.9) | 0.08 (0.02–0.32) | **<0.0001** | 0.09 (0.02–0.46) | **0.003** | 7 (3.4) | 0.15 (0.05–0.44) | **<0.0001** | 0.24 (0.07–0.81) | **0.021** |
| **STI pathogens** | | | | | | | | | | | | |
| U. parvum | 402 (42.6) | 230 (39.8) | 74 (46.5) | 1.32 (0.93–1.88) | 0.126 | 1.39 (0.92–2.11) | 0.116 | 98 (47.3) | 1.36 (0.99–1.87) | 0.059 | 1.06 (0.73–1.55) | 0.762 |
| C. trachomatis | 145 (15.4) | 45 (7.8) | 35 (22.0) | 3.34 (2.06–5.42) | **<0.0001** | 2.78 (1.60–4.83) | **<0.0001** | 65 (31.4) | 5.42 (3.55–8.27) | **<0.0001** | 5.09 (3.17–8.17) | **<0.0001** |
| M. hominis | 120 (12.7) | 57 (9.9) | 20 (12.6) | 1.32 (0.76–2.26) | 0.321 | 1.15 (0.62–2.13) | 0.669 | 43 (20.8) | 2.40 (1.55–3.70) | **<0.0001** | 2.03 (1.22–3.39) | **0.006** |
| K. granulomatis | 102 (10.8) | 48 (8.3) | 24 (15.1) | 1.96 (1.16–3.32) | **0.011** | 2.40 (1.34–4.30) | **0.003** | 30 (14.5) | 1.87 (1.15–3.05) | **0.011** | 2.36 (1.38–4.04) | **0.002** |
| U. urealyticum | 73 (7.7) | 39 (6.7) | 13 (8.2) | 1.23 (0.64–2.37) | 0.533 | 1.02 (0.48–2.18) | 0.958 | 21 (10.1) | 1.56 (0.90–2.72) | 0.114 | 1.25 (0.66–2.39) | 0.494 |
| T. pallidum | 32 (3.4) | 3 (0.5) | 12 (7.5) | 15.65 (4.36–56.17) | **<0.0001** | 19.76 (5.19–75.13) | **<0.0001** | 17 (8.2) | 17.15 (4.97–59.16) | **<0.0001** | 23.79 (6.54–86.48) | **<0.0001** |
| M. genitalium | 18 (1.9) | 10 (1.7) | 4 (2.5) | 1.47 (0.45–4.74) | 0.515 | 1.25 (0.35–4.55) | 0.731 | 4 (1.9) | 1.12 (0.35–3.61) | 0.768 | 0.46 (0.11–1.84) | 0.269 |
| N. gonorrhoeae | 17 (1.8) | 2 (0.3) | 9 (5.7) | 17.28 (3.70–80.82) | **<0.0001** | 28.11 (5.54–142.47) | **<0.0001** | 6 (2.9) | 8.60 (1.72–42.94) | **0.005** | 7.20 (1.33–38.93) | **0.022** |
| H. ducreyi | 5 (0.5) | 1 (0.2) | 2 (1.3) | 7.35 (0.66–81.59) | 0.119 | 12.27 (1.05–143.59) | **0.046** | 2 (1.0) | 5.63 (0.51–62.41) | 0.172 | 8.13 (0.68–97.21) | 0.098 |
| HCMV | 228 (24.2) | 127 (22.0) | 39 (24.5) | 1.15 (0.77–1.74) | 0.494 | 0.96 (0.60–1.54) | 0.876 | 62 (30.0) | 1.52 (1.06–2.17) | **0.021** | 1.57 (1.05–2.34) | **0.029** |
| HPV | 148 (15.7) | 61 (10.6) | 45 (28.3) | 3.35 (2.16–5.17) | **<0.0001** | 2.43 (1.47–4.00) | **0.001** | 42 (20.3) | 2.16 (1.40–3.32) | **<0.0001** | 1.20 (0.73–2.09) | 0.433 |
| HHV-8 | 117 (12.4) | 67 (11.6) | 16 (10.1) | 0.85 (0.48–1.52) | 0.589 | 0.82 (0.43–1.58) | 0.558 | 34 (16.4) | 1.50 (0.96–2.34) | 0.075 | 1.79 (1.09–2.95) | **0.023** |
| VZV | 23 (2.4) | 9 (1.6) | 6 (3.8) | 2.48 (0.87–7.07) | 0.107 | 2.42 (0.73–8.00) | 0.147 | 8 (3.9) | 2.54 (0.97–6.68) | 0.089 | 1.83 (0.59–5.67) | 0.294 |
| HSV-1 | 15 (1.6) | 11 (1.9) | 2 (1.3) | 0.66 (0.14–2.99) | 0.745 | 0.82 (0.16–4.10) | 0.810 | 2 (1.0) | 0.50 (0.11–2.29) | 0.531 | 0.78 (0.16–3.74) | 0.758 |
| HSV-2 | 3 (0.3) | 3 (0.5) | 0 (0) | | 0.999 | | | 0 (0) | | 0.570 | | |
| T. vaginalis | 17 (1.8) | 7 (1.2) | 6 (3.8) | 3.20 (1.06–9.66) | **0.041** | 3.30 (0.88–12.44) | 0.078 | 4 (1.9) | 1.61 (0.47–5.55) | 0.493 | 2.51 (0.60–10.52) | 0.209 |

[a] ORs and 95% CIs were calculated by univariable analysis.
[b] Chi-square test and Fisher's exact test were used to compare variables between groups. Bold indicates significance (P < 0.05).
[c] Adjustment was made in a multivariable model stratified by cohort for age (five categories).
[d] P value for the multivariable model. Bold indicates significance (P < 0.05).

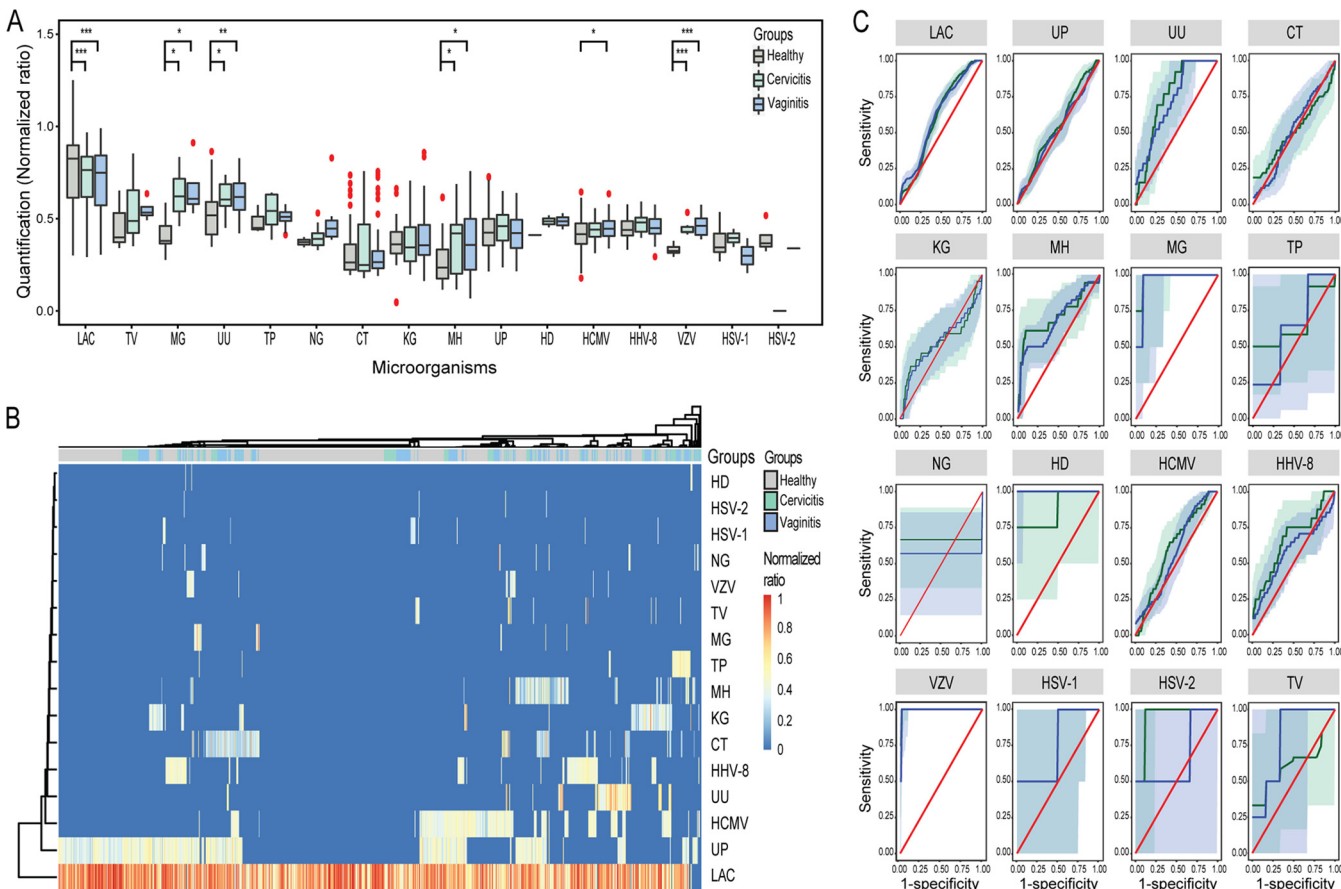

**FIG 3** Normalized abundance of STI pathogens and *Lactobacillus* spp. in healthy subjects and those with cervicitis and vaginitis. (A) Box plot showing comparison of normalized abundance ratio of microbial load to human RNase P load obtained from qPCR. The bottom and top bars of the boxes represent the interquartile range (2.5th and 97.5th percentiles), and the lines within the boxes denote the median value. Circles represent data points beyond the whiskers. *, $P < 0.05$; **, $P < 0.001$; ***, $P < 0.0001$. (B) Heat map of 944 samples from three study groups. The normalized abundance ratio is illustrated by the color key. Each column represents a subject. Columns are clustered using normalized abundance ratio and color coded by groups: healthy, cervicitis, and vaginitis. Clustering of each microorganism was generated using unweighted pair-group method with arithmetic mean (UPGMA) clustering with Euclidean distance. (C) ROC curve showing the area under the ROC curves (AUC) of each microorganism detected in cervicitis and vaginitis. A cutoff was calculated to maximize discrimination between cases and controls: using all cases and controls in which each microorganism was detected, the maximum Youden index (YI) from the ROC curve with case-control status as the outcome and microbial normalized abundance ratio as the independent variable was determined, where YI = sensitivity + specificity − 1. LAC, *Lactobacillus* spp.; other abbreviations are as in Fig. 2.

*T. pallidum*, *K. granulomatis*, *M. hominis*, and *N. gonorrhoeae*; quantitatively for *C. trachomatis*, *U. parvum*, HCMV, *M. hominis*, *T. pallidum*, HHV-8, VZV, *T. vaginalis*, *U. urealyticum*, *K. granulomatis*, and *N. gonorrhoeae*). Five pathogens, including *C. trachomatis*, *T. pallidum*, *M. hominis*, *K. granulomatis*, and *N. gonorrhoeae*, were present both qualitatively and quantitatively. The diagnostic performance of the random forest classification was assessed using the validation cohort. The results demonstrated that the random classifier works best on incorporating both qualitative and quantitative data sets, and the selected 18 features achieved an accuracy of 0.96 (95% CI, 0.93 to 0.98) for cervicitis and 0.86 (95% CI, 0.82 to 0.90) for vaginitis. The sensitivity, specificity, and AUC were 0.92, 0.98, and 0.98 (95% CI, 0.96 to 0.99), respectively, for cervicitis and 0.70, 0.92, and 0.89 (95% CI, 0.86 to 0.92), respectively, for vaginitis (Fig. 5B and C).

## DISCUSSION

In this study, we systematically explored the association of STI pathogens with cervicitis and vaginitis. Qualitative and quantitative analyses displayed the odds ratio for each pathogen, while logistic regression and random forest analysis disclosed the collective associations of both the presence and abundance of the pathogens with the conditions. Consistency among the four tiers supported the etiological roles observed

**TABLE 2** Detection of microorganism in healthy, cervicitis, and vaginitis groups based on the cutoff values of the NAR

| Microorganism | Healthy controls vs cervicitis patients | | | | | Healthy controls vs vaginitis patients | | | | |
|---|---|---|---|---|---|---|---|---|---|---|
| | Cutoff (NAR)[a] | No. (%) of: | | OR (95% CI)[b] | P[c] | Cutoff (NAR)[a] | No. (%) of: | | OR (95% CI)[b] | P[c] |
| | | Healthy controls | Cervicitis patients | | | | Healthy controls | Vaginitis patients | | |
| Lactobacillus spp. | >0.84 | 250 (43.3) | 31 (19.5) | 0.32 (0.21–0.49) | **<0.0001** | >0.79 | 332 (57.4) | 72 (34.8) | 0.40 (0.28–0.55) | **<0.0001** |
| | ≤0.84 | 326 (56.4) | 118 (74.2) | 2.22 (1.50–3.29) | **0.0001** | ≤0.79 | 244 (42.2) | 127 (61.4) | 2.17 (1.57–3.01) | **<0.0001** |
| T. vaginalis | >0.39 | 3 (0.5) | 5 (3.1) | 6.22 (1.47–26.33) | **0.014** | >0.41 | 2 (0.3) | 4 (1.9) | 5.68 (1.03–31.22) | **0.045** |
| | ≤0.39 | 3 (0.5) | 1 (0.6) | 1.21 (0.13–11.74) | 0.868 | ≤0.41 | 4 (0.7) | 0 (0) | | 0.430 |
| M. genitalium | >0.46 | 3 (0.5) | 4 (2.5) | 4.95 (1.10–22.33) | **0.042** | >0.47 | 2 (0.3) | 4 (1.9) | 5.67 (1.03–31.22) | **0.045** |
| | ≤0.46 | 9 (1.6) | 0 (0) | | 0.250 | ≤0.47 | 10 (1.7) | 0 (0) | | 0.160 |
| U. urealyticum | >0.55 | 14 (2.4) | 10 (6.3) | 2.70 (1.18–6.21) | **0.022** | >0.55 | 14 (2.4) | 16 (7.7) | 3.37 (1.62–7.04) | **0.001** |
| | ≤0.55 | 27 (4.7) | 3 (1.9) | 0.39 (0.12–1.31) | 0.129 | ≤0.55 | 27 (4.7) | 5 (2.4) | 0.51 (0.19–1.33) | 0.167 |
| T. pallidum | >0.57 | 0 (0) | 6 (3.8) | | **<0.0001** | >0.45 | 1 (0.2) | 14 (6.8) | 41.85 (5.47–320.39) | **<0.0001** |
| | ≤0.57 | 3 (0.5) | 6 (3.8) | 7.52 (1.86–30.40) | **0.005** | ≤0.45 | 2 (0.3) | 3 (1.4) | 4.24 (0.70–25.53) | 0.115 |
| N. gonorrhoeae | >0.40 | 0 (0) | 4 (2.5) | | **0.002** | >0.40 | 0 (0) | 5 (2.4) | | **0.001** |
| | ≤0.40 | 2 (0.3) | 5 (3.1) | 9.35 (1.80–48.66) | **0.008** | ≤0.40 | 2 (0.3) | 2 (1.0) | 2.81 (0.39–20.08) | 0.303 |
| C. trachomatis | >0.65 | 2 (0.3) | 6 (3.8) | 11.29 (2.26–56.51) | **0.002** | >0.23 | 28 (4.8) | 47 (22.7) | 5.77 (3.50–9.51) | **<0.0001** |
| | ≤0.65 | 40 (6.9) | 26 (16.4) | 2.63 (1.55–4.46) | **0.0003** | ≤0.23 | 14 (2.4) | 14 (6.8) | 2.92 (1.37–6.24) | **0.006** |
| K. granulomatis | >0.45 | 7 (1.2) | 6 (3.8) | 3.20 (1.06–9.66) | **0.041** | >0.45 | 6 (1.0) | 9 (4.3) | 4.33 (1.52–12.33) | **0.006** |
| | ≤0.45 | 40 (6.9) | 16 (10.1) | 1.50 (0.82–2.77) | 0.188 | ≤0.45 | 41 (7.1) | 21 (10.1) | 1.48 (0.85–2.57) | 0.165 |
| M. hominis | >0.41 | 5 (0.9) | 10 (6.3) | 7.69 (2.59–22.84) | **<0.0001** | >0.41 | 5 (0.9) | 19 (9.2) | 11.58 (4.27–31.45) | **<0.0001** |
| | ≤0.41 | 45 (7.8) | 8 (5.0) | 0.63 (0.29–1.36) | 0.238 | ≤0.41 | 45 (7.8) | 21 (10.1) | 1.34 (0.78–2.30) | 0.295 |
| U. parvum | >0.47 | 77 (13.3) | 37 (23.3) | 1.97 (1.27–3.06) | **0.002** | >0.30 | 207 (35.8) | 91 (44.0) | 1.41 (1.02–1.94) | **0.038** |
| | ≤0.47 | 158 (27.3) | 40 (25.2) | 0.89 (0.60–1.34) | 0.583 | ≤0.30 | 28 (4.8) | 13 (6.3) | 1.32 (0.67–2.59) | 0.427 |
| H. ducreyi | >0.41 | 0 (0) | 2 (1.3) | | **0.046** | >0.41 | 0 (0) | 2 (1.0) | | 0.069 |
| | ≤0.41 | 1 (0.2) | 0 (0) | | 0.909 | ≤0.41 | 1 (0.2) | 0 (0) | | 0.963 |
| HCMV | >0.40 | 70 (12.1) | 27 (17.0) | 1.48 (0.92–2.41) | 0.108 | >0.38 | 81 (14.0) | 54 (26.1) | 2.17 (1.47–3.20) | **<0.0001** |
| | ≤0.40 | 53 (9.2) | 7 (4.4) | 0.46 (0.20–1.02) | 0.057 | ≤0.38 | 42 (7.3) | 6 (2.9) | 0.38 (0.16–0.91) | **0.030** |
| HHV-8 | >0.47 | 22 (3.8) | 10 (6.3) | 1.70 (0.79–3.66) | 0.174 | >0.43 | 32 (5.5) | 25 (12.1) | 2.34 (1.35–4.06) | **0.002** |
| | ≤0.47 | 38 (6.6) | 6 (3.8) | 0.56 (0.23–1.34) | 0.193 | ≤0.43 | 28 (4.8) | 9 (4.3) | 0.89 (0.41–1.93) | 0.773 |
| VZV | >0.38 | 0 (0) | 6 (3.8) | | **<0.0001** | >0.38 | 0 (0) | 8 (3.9) | | **<0.0001** |
| | ≤0.38 | 11 (1.9) | 0 (0) | | 0.197 | ≤0.38 | 11 (1.9) | 0 (0) | | 0.141 |
| HSV-1 | >0.33 | 6 (1.0) | 2 (1.3) | 1.21 (0.24–6.08) | 0.685 | >0.21 | 12 (2.1) | 1 (0.5) | 0.23 (0.03–1.77) | 0.202 |
| | ≤0.33 | 6 (1.0) | 0 (0) | | 0.381 | ≤0.21 | 0 (0) | 1 (0.5) | | 0.963 |
| HSV-2 | >0.32 | 2 (0.3) | 0 (0) | | 1.000 | >0.32 | 2 (0.3) | 1 (0.5) | 1.40 (0.13–15.50) | 1.000 |
| | ≤0.32 | 1 (0.2) | 0 (0) | | 0.909 | ≤0.32 | 1 (0.2) | 0 (0) | | 0.963 |

[a]For each microorganism, cutoff values were set according to the Youden index using MedCalc. The cutoff was categorized using the NAR of microbial load ($log_{10}$ copies/mL/human RNase P load ($log_{10}$ copies/mL). The cutoff is the predicted probability of the logistic regression model, where 0 = healthy and 1 = cervicitis or vaginitis.
[b]Odds ratios and 95% CIs were calculated by univariable logistic regression analysis.
[c]Chi-square test and Fisher's exact test were used to compare variables between groups. Bold indicates significance ($P < 0.05$).

**TABLE 3** Adjusted odds ratios showing the association between the NAR of the microorganisms above the cutoff and women with cervicitis and vaginitis

| Variable | Multivariable analysis based on NAR above the cutoff[a] | | | |
|---|---|---|---|---|
| | Cervicitis | | Vaginitis | |
| | aOR[b] (95% CI) | P[c] | aOR[b] (95% CI) | P[c] |
| Age (yrs) | | | | |
| 16–25 | Reference | | Reference | |
| 26–35 | 0.20 (0.09–0.43) | **<0.0001** | 0.32 (0.14–0.71) | **0.005** |
| 36–45 | 0.21 (0.09–0.47) | **<0.0001** | 0.31 (0.14–0.71) | **0.005** |
| 46–55 | 0.14 (0.06–0.35) | **<0.0001** | 0.20 (0.08–0.49) | **<0.0001** |
| ≥56 | 0.06 (0.01–0.32) | **0.001** | 0.12 (0.03–0.51) | **0.004** |
| Microorganisms | | | | |
| *Lactobacillus* spp. | 0.36 (0.23–0.58) | **<0.0001** | 0.44 (0.29–0.65) | **<0.0001** |
| *T. vaginalis* | 3.25 (0.50–21.16) | 0.217 | 5.74 (0.96–34.24) | 0.055 |
| *M. genitalium* | 8.42 (1.53–46.45) | **0.015** | 1.79 (0.21–15.48) | 0.597 |
| *U. urealyticum* | 2.90 (1.05–8.00) | **0.040** | 2.91 (1.10–7.70) | **0.032** |
| *T. pallidum* | | | 43.23 (5.33–350.67) | **<0.0001** |
| *C. trachomatis* | 8.74 (1.29–59.11) | **0.026** | 4.42 (2.49–7.83) | **<0.0001** |
| *K. granulomatis* | 1.90 (0.50–7.28) | 0.347 | 4.65 (1.37–15.82) | **0.014** |
| *M. hominis* | 6.29 (1.82–21.72) | **0.004** | 10.40 (3.37–32.09) | **<0.0001** |
| *U. parvum* | 2.45 (1.47–34.07) | **0.001** | 1.44 (0.96–2.15) | 0.077 |
| HCMV | 1.03 (0.58–1.82) | 0.920 | 1.55 (0.97–2.47) | 0.069 |
| HHV-8 | 1.90 (0.81–4.47) | 0.139 | 2.53 (1.32–4.83) | **0.005** |
| HSV-1 | 2.30 (0.44–12.13) | 0.326 | 0.43 (0.05–3.46) | 0.430 |
| HSV-2 | | | 0.13 (0.01–3.63) | 0.232 |

[a]Cutoff values were set according to the Youden index, calculated using MedCalc.
[b]Adjustment was made in a multivariable model stratified by cohort for age (five categories).
[c]Bold indicates significance ($P < 0.05$).

in the two conditions. The established models not only form a foundation for diagnosis of the conditions but also help understand their pathological difference.

One finding is that STI pathogens play a major role in cervicitis but a lesser one in vaginitis. This could be seen in the larger AUC of the former than the latter from either logistic regression (0.85 versus 0.72) or random forest (0.98 versus 0.89) models when assessed with the validation groups. This result is in line with the mechanisms of cervicitis and vaginitis: cervicitis is mainly caused by STIs, whereas vaginitis is an imbalance in the microbiota (17). As seen from the features selected for each condition, cervicitis is more associated with qualitative features, whereas vaginitis is more associated with quantitative features. These data reflect the fact that the former is caused by infection with STI pathogens, while the latter is caused by their higher load.

HPV, which causes genital warts and cervical cancer, was seen as an exceptional feature of cervicitis by both logistic regression and random forest analysis. This result is in line with evidence from a previous study that demonstrated a higher detection rate of HPV among patients with mucopurulent cervicitis (18). Another example is *M. genitalium*, which was seen as a quantitative feature for cervicitis only. This finding also agreed with reports showing a high prevalence of *M. genitalium* in women with cervicitis (19, 20). Conversely, HCMV, HHV-8, and *T. vaginalis* were seen as vaginitis features in terms of their microbial loads. Of these pathogens, *T. vaginalis* is a well-established trichomonas vaginitis (trichomoniasis)-causing pathogen. Regarding HCMV and HHV-8, despite the paucity of data regarding their direct association with vaginitis, studies showed that vaginitis can affect either the initial infection by HCMV or virus replication/shedding (21). For instance, HCMV DNA has been more frequently detected in vaginal washings from women with bacterial vaginosis than in women with normal genital tract flora.

Cervicitis and vaginitis are closely related conditions, as the former is essentially an upper genital tract inflammation and the latter is a lower genital tract inflammation. The anatomical proximity of the cervix and vagina might facilitate pathogen transmission between them. Thus, certain pathogens can be present in both conditions. This relationship was

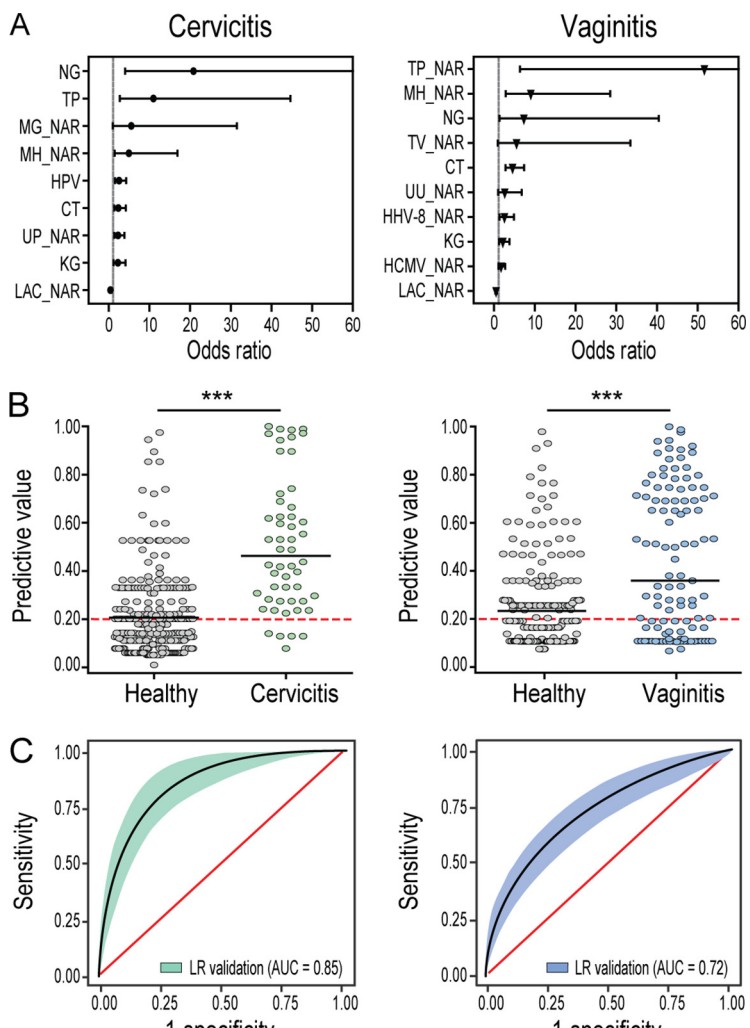

**FIG 4** Classification variables selected by logistic regression models. (A) Forest plot representing the odds ratio and 95% confidence interval for the association of significant microbial agents present in cervicitis and vaginitis. Each microbial agent was analyzed using logistic regression in a multivariable model. This plot shows the significant association for combined qualitative and quantitative screening of microbial agents with cervicitis and vaginitis after adjustment for potential confounding factor. Adjusted odds ratios with 95% confidence intervals are shown in black, with odds ratios illustrated by circles. (B) Predictive value of the logistic regression models for patients and healthy subjects in the validation data set. Scatterplots showing the predictive value of the models for distinguishing cervicitis patients versus healthy controls and vaginitis patients versus healthy controls. Horizontal lines indicate the medians, and red dotted lines show the cutoff value of 0.20. ***, $P < 0.0001$ (Mann-Whitney $U$ test). (C) Area under the ROC curves (AUC) of qualitative, quantitative, and simultaneous qualitative and quantitative PCR combined screening of STI pathogens discriminates women with cervicitis (left panel) and vaginitis (right panel) from healthy subjects.

reflected in STI pathogens shared by both conditions. For example, *C. trachomatis* and *N. gonorrhoeae* are the two primary etiological agents of cervicitis and were listed as important features of cervicitis. Of note, *C. trachomatis* was listed as an important vaginitis-causing agent. This was also true for *N. gonorrhoeae*, though with less importance for vaginitis. Similarly, *K. granulomatis*, *M. hominis* NAR, and *Lactobacillus* spp. NAR were all shared by the two conditions. *K. granulomatis* causes donovanosis, also known as granuloma inguinale, through sexual contact. If left unchecked, sores caused by the infection can fuel the spread of HIV (22). As the common sites where women are affected are the labia minora, fourchette, cervix, and upper genital tract, *K. granulomatis* can be a causative agent of both cervicitis and vaginitis. *M. hominis* is reportedly present in almost everyone's urinary tract in small quantities, but higher quantities can cause urethritis and

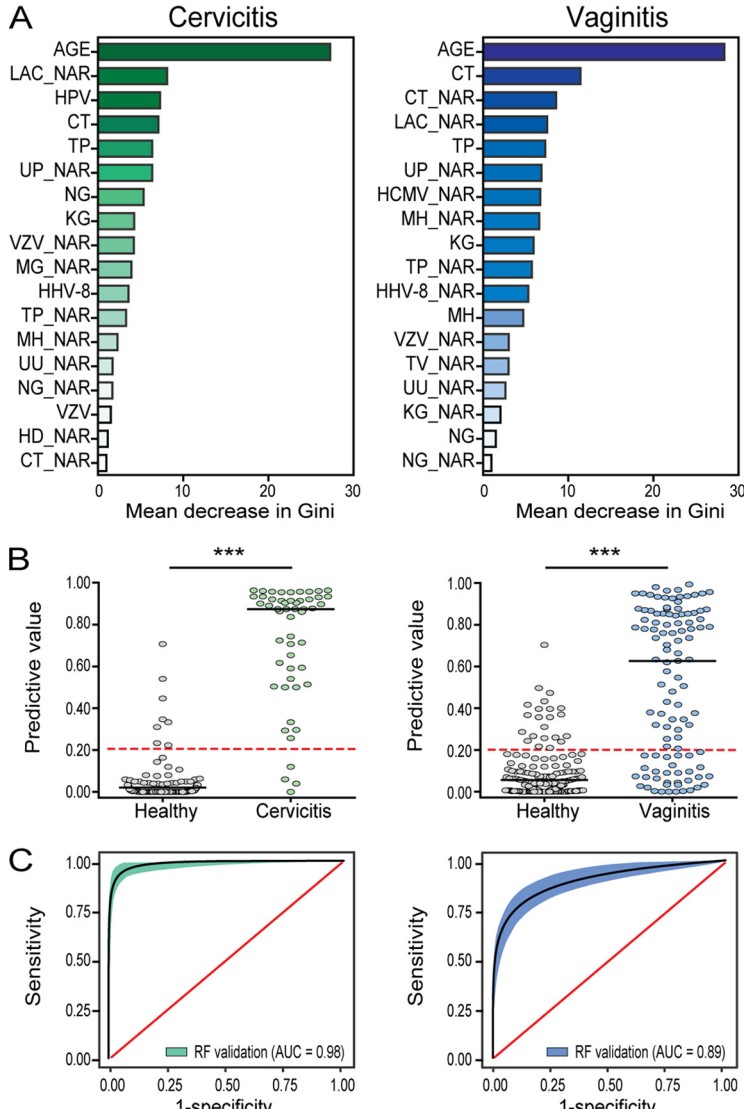

**FIG 5** The classification features selected by random forest models. (A) Random forest ranking plot of variable importance in predicting risk of cervicitis and vaginitis. The *x* axis shows the importance of the feature to the accuracy of the model, which was estimated by calculating the mean decrease in Gini after randomly permuting the values of each given feature. (B) Predictive value of the random forest models for distinguishing patients and healthy controls in the validation data set. Scatterplots show the predictive value of the random forest models for cervicitis and vaginitis. Horizontal lines indicate the medians, and red dotted lines show the cutoff value of 0.20. ***, *P* < 0.0001 (Mann-Whitney *U* test). (C) Area under the ROC curves (AUC) curve of the random forest classification models for cervicitis versus healthy subjects (left) and vaginitis versus healthy subjects (right).

increase the risk of vaginitis and pelvic inflammatory disease in women through sexual transmission (23). This explained the importance of quantitative detection of *M. hominis*. As the symptoms of *M. hominis* infection are similar to those of many other STIs and since the condition can often be mistaken for gonorrhea or *chlamydia*, its identification is of great importance for appropriate treatment. It is well known that high abundance of *Lactobacillus* spp. in the vagina is considered "healthy" or "normal" and low abundance or absence is considered an "abnormal" condition, as in vaginitis. However, studies also showed that the absence of $H_2O_2$-producing lactobacilli may contribute to the development of cervicitis (24).

Because random forest models include more features than the logistic regression models, certain pathogens classified to one condition by logistic regression could be included in another condition by the random forest models, and vice versa. For example, *T. pallidum*

and *U. parvum* NAR were classified as cervicitis variables while *T. pallidum* NAR and *U. urealyticum* NAR were classified as vaginitis variables in logistic regression models. All were, however, classified as important features for both cervicitis and vaginitis in the random forest models (Fig. S4). When examined individually, *T. pallidum* was listed as the second most important variable for cervicitis and *T. pallidum* NAR was the top variable for vaginitis in the logistic regression models. In the random forest models, *T. pallidum* was listed as the fifth important feature for both cervicitis and vaginitis. These results demonstrate the critical role of *T. pallidum*, a highly invasive STI pathogen, in these two conditions regardless of the loads. Unlike *T. pallidum*, both *U. parvum* and *U. urealyticum* were selected as quantitative features for the two conditions, implying that both of these agents can be etiological pathogens only when present at high loads. *Ureaplasma* spp. have been regarded as commensals in the genital tracts of healthy women, and their presence was predictive only when measured quantitatively (e.g., at high abundance). This result was concordant with a previous observation that high density of *U. parvum* and *U. urealyticum* is associated with cervicitis (25). So far, *U. parvum* and *U. urealyticum* are mostly detected qualitatively. Consensus on their association with cervicitis and vaginitis has yet to be achieved. Our findings support the necessity of quantitative detection of *U. parvum* and *U. urealyticum* in these conditions.

Extra features found in the random forest model include VZV, HHV-8, and *H. ducreyi* NAR for cervicitis only, *M. hominis* and *K. granulomatis* NAR for vaginitis only, and *C. trachomatis* NAR, *N. gonorrhoeae* NAR, and VZV NAR for both. Among these features, *H. ducreyi* NAR for cervicitis, *K. granulomatis* NAR for vaginitis, *C. trachomatis* NAR for cervicitis, and *N. gonorrhoeae* NAR for both are all features of low importance in the models. VZV is also of less importance for cervicitis but VZV NAR is more important for vaginitis according to their ranks, indicating that a high load of VZV is associated with vaginitis. The surprisingly higher prevalence of VZV than the other two viruses of the same family (*Herpesviridae*), HSV-1 and HSV-2 (which are not listed as features for any conditions due to their rarity in this study), indicates its new predictive value in vaginitis. The fact that HHV-8 NAR is associated with vaginitis strongly suggested the importance of the high load in vaginitis. Unlike HHV-8, both *M. hominis* and *M. hominis* NAR were selected features for vaginitis, with the latter being more important than the former. Since *M. hominis* NAR is also listed as a feature of cervicitis, we conclude that a high load of *M. hominis* is strongly associated with both vaginitis and cervicitis. Interestingly, *C. trachomatis* NAR was listed as the third important feature after *C. trachomatis* for vaginitis. *C. trachomatis* NAR was, however, listed as the least important feature for cervicitis despite *C. trachomatis* itself being listed as the fourth important feature. These results imply that higher load of *C. trachomatis* is mainly associated with vaginitis but not with cervicitis. Collectively, these observations support the idea that vaginitis is more associated with quantitative features than cervicitis.

A direct application of knowing the roles of the STI pathogens in these conditions is establishing pathogen-based diagnostic assays. This is particularly true for cervicitis, which showed a sensitivity of 92% and a specificity of 98% in the validation sample of the random forest model. Of the 18 features selected by the random forest model, age could be easily obtained from the patient or through case reporting systems. The HPV status could be gained through available HPV genotyping assays. The other 16 features involved qualitative and/or quantitative information of 11 STI pathogens and *Lactobacillus* spp. Such information could be obtained through either a multiplex qPCR assay for quantitative detection or an abbreviated assay containing an 11-plex MeltArray for screening followed by singleplex qPCR assays for quantification of the positive targets. Diagnosis of cervicitis is currently symptom-based with assistance from microscopy. Pathogen-based diagnosis is often limited to *C. trachomatis* and *N. gonorrhoeae*, leaving half of patients undiagnosed with respect to pathogen types (26). Although assays for different STI pathogens are available, pathologists often hesitate to use these assays without knowing the etiological roles of these organisms, leaving clinicians unsure how to treat patients. This situation could be immediately improved by the pathogen-based assays described in the present work.

Unlike cervicitis, vaginitis has already been diagnosed through microbe-based assays for various types, including bacterial vaginosis, candida vaginitis (vaginal candidiasis), and trichomonas vaginitis (trichomoniasis) (27). However, vaginitis is a complex gynecological inflammation, the detection method adopted must consider both endogenous flora and foreign pathogens, involving a large variety of microorganisms. Meanwhile, the detection of the microecological flora requires both qualitative detection of its composition and quantitative detection of its abundance, which poses a serious challenge to the existing detection schemes. Also, there are still unknown causative microorganisms, for instance, in aerobic vaginitis, which is characterized by an abnormal vaginal microflora containing aerobic enteric bacteria, and the exact types and numbers of the members of the microflora in aerobic vaginitis remain largely undefined (28). In this regard, inclusion of STI pathogens can be a useful complement to current assays for diagnosis of vaginitis. The 18-feature random forest model involving 11 STI pathogens and *Lactobacillus* spp. produced an AUC of 0.89 (95% CI, 0.86 to 0.92), corresponding to a sensitivity of 70% and specificity of 92% in the validation sample of the random forest model. The lower sensitivity for vaginitis than cervicitis (70% versus 92%) reflects the fact that our models were limited to STI pathogens despite the fact that many non-STI microbes can cause vaginitis. Based on these results, a vaginitis-oriented STI pathogen assay could be established. The addition of such an STI pathogen-based assay to vaginitis diagnosis could help identify this segment of vaginitis patients being missed by current assays.

Diagnosis of infection has been hindered by not knowing the etiological role of the pathogens. This is particularly true for STIs, which involve a large number of microorganisms with frequent coinfections (29, 30). Most current STI assays are restricted to a limited number of conditions caused by a single pathogen, including *chlamydia*, gonorrhea, syphilis, and trichomonas. Remarkable advances in new diagnostic point-of-care tests have been made in recent years (31). However, their pragmatic improvement for the control of STIs is still compromised due to their single-target design. Thus far, laboratory etiological diagnosis, which is considered the gold standard, remains difficult for STIs. Rapidly identifying the exact disease-causing pathogen(s) would enable appropriate drug selection, prognosis, progress monitoring, and even prevention planning, thus leading to greatly improved patient management for STIs. It would be intriguing to see whether inclusion of the STI pathogens associated with cervicitis and vaginitis could enhance the diagnosis performance.

This study has several limitations. First, all participants for model-building were from one hospital; additional multicenter studies with larger cohorts are essential for validation of our models. Second, our models were based on a retrospective cohort that may have some level of bias. A prospective cohort is needed to evaluate the clinical applicability. Third, the pathogens included in our study are all from the MeltArray STI assay. Although many of them are proved, classical STI-causing pathogens, some (e.g., VZV and HHV-8), although they have been implicated in some STIs (32, 33), remained to be confirmed. Their exact roles in STIs needs further study. Also, our study did not explore other microbes of the vaginal microbiota, including aerobic vaginitis- and bacterial vaginosis-associated microbes and other potential STI pathogens. Taking these additional microbiome biomarkers into account might enable the development of more robust predictive models that could be used for risk stratification to guide clinical decision-making. Last, the risk for both cervicitis and vaginitis involves many behavioral factors, such as number of sexual partners, vaginal douching, a history of vaginal infections, etc., which were not assessed in the present study. It would be preferable to include these factors to improve diagnostic performance and refine the conclusions.

Overall, our findings suggest that STI pathogens can be useful microbiological predictive markers of both cervicitis and vaginitis. Simultaneous qualitative and quantitative detection of a wide range of STI pathogens and further coupling with machine learning models can be useful in establishing pathogen-based diagnosis of gynecological inflammation. The high prevalence and increased risk of STI pathogens in women with cervicitis and vaginitis provides a durable stimulus for comprehensive testing of these

pathogens. Pathogen-based diagnosis should be of value in guiding the rational use of antimicrobials to control the spread of STIs.

## MATERIALS AND METHODS

**Study design and study populations.** An initial collection of 203 vaginal swab samples was used to evaluate the concordance of the qPCR assay and MeltArray assay in the identification of the STI pathogens. These samples were obtained from Liuzhou Maternal and Child Health Hospital and the First Affiliated Hospital of Xiamen University and were all deidentified when received. A second collection of vaginal or cervical samples were from 1,453 women, aged 16 or older, presenting at the maternity and gynecological clinic of Xiamen Maternal and Child Health Hospital between March 2019 to February 2020. Women who were admitted with either cervicitis or vaginitis symptoms were recruited as patients. Women who underwent annual gynecologic examination were recruited as controls. Exclusion criteria included menstruation, pregnancy, and recent use of hormones and antibiotics. The clinical data were extracted from the electronic medical records. The entire population was divided into a derivation cohort recruited from March 2019 to August 2019 and a validation cohort recruited from September 2019 to February 2020.

**Molecular assays.** A 16-plex MeltArray STI assay (Zeesan Biotech, Xiamen, China) was used to screen for 15 STI pathogens according to the manufacturer's instructions. Briefly, PCR and melting curve analysis were carried out on a SLAN-96S real-time PCR instrument (Hongshi Medical Technology, Shanghai, China). The program was as follows: denaturation at 95℃ for 5 min; 50 cycles of 95℃ for 20 s and 60℃ for 1 min; and then 35℃ for 30 min, 95℃ for 2 min, 45℃ for 2 min, followed by a temperature increase from 45℃ to 95℃ (0.04℃/step). Fluorescence intensity was measured in four detection channels (6-carboxyfluorescein [FAM], 6-carboxy-2,4,4,5,7,7-hexachlorofluorescein [HEX], carboxyrhodamine [ROX], and Cy5) at each step during melting curve analysis. The $T_m$ values were obtained by identification of the peaks of the melting curves (Table S6). The human RNase P gene was used as an internal positive control. HPV genotyping was performed for the identification of 37 HPV genotypes using a commercial HPV GenoArray Diagnostic kit (HybriBio, Ltd., Guangdong, China) according to the manufacturer's instructions. The HPV genotypes include 19 high-risk genotypes (HPV 16, 18, 26, 31, 33, 34, 35, 39, 45, 51, 52, 53, 56, 58, 59, 66, 68, 73, and 82) and 18 low-risk genotypes (HPV 6, 11, 40, 42, 43, 44, 54, 55, 57, 61, 67, 69, 70, 71, 72, 81, 83, and 84).

**Singleplex qPCR assays.** Singleplex species-specific qPCR assays were established according to the published literature. These assays were performed individually to quantify the 15 STI pathogens mentioned above as well as *Lactobacillus* spp. and human RNase P. Dominance of *Lactobacillus* spp. is regarded as an indicator of a healthy vagina (34). Human RNase P is used as an internal positive control to normalize the quantity of pathogen in the swab sample as well as to indicate the success of sample collection. The primer and probes used are listed in Table S7. PCR was run on a Bio-Rad CFX96 PCR system (Bio-Rad, Hercules, CA, USA). To determine the LOD for each target, 10-fold serially diluted plasmid DNA templates, ranging from $1.0 \times 10^7$ copies/$\mu$L to $1.0 \times 10^0$ copy/$\mu$L, were initially detected in triplicate for each dilution to obtained an approximated LOD. Confirmation of the LOD was further achieved by testing 20 replicates of a target at its provisional LOD. The LOD was defined as the dilution of the lowest concentration that can be detected in no less than 95% of 20 replicate tests. The absolute quantity (copy number) of the target genomic DNA in the sample was determined according to a calibration curve constructed within the quantification range ($10^1$ to $10^7$ copies/$\mu$L) with a minimum coefficient of determination ($R^2$) greater than 0.99 (Table S1). The plasmids were each artificially synthesized to encompass the target sequence for PCR. The concentration of plasmid DNA extracted from an engineering *Escherichia coli* strain (DH5$\alpha$) was measured using a NanoDrop ND-2000 spectrophotometer (Thermo Fisher Scientific, CA, USA). The normalized abundance ratio (NAR) was calculated from the absolute counts of microbial copy numbers relative to the human RNase P copy number, as follows (35): microorganism load [log10(copies/mL)]/human RNase P load [log10(copies/mL)].

**Sample collection and clinical diagnosis.** Vaginal or cervical specimens were collected by a gynecological practitioner on the day of the clinical visit. For cervical sample collection, a specimen was taken from the cervical canal using a specific cervical brush (HybriBio Ltd., Guangdong, China) after cleaning the cervix with a cotton swab. For vaginal sample collection, two swabs were placed into the vagina at the standard anatomical site (lateral vaginal wall). The first vaginal swab was processed for Gram staining, and the second one was used for genomic DNA extraction. Cervicitis was defined clinically by observation of mucopurulent discharge at the cervix or at least 30 polymorphonuclear leukocytes per high-power magnification field on endocervical Gram staining. Vaginitis was diagnosed clinically based on Nugent's scoring system. A microscopy score of 0 to 10 was assigned by an experienced microbiologist according to the standardized method described by Nugent et al. (36). Additionally, diagnosis of vaginitis was also done by Amsel's criteria (37) for patients with relevant clinical symptoms.

**Pretreatment of clinical samples.** All samples were coded and stored at −20℃ before use. DNA extraction was carried out with a Lab-Aid 824s automated DNA extraction system (Zeesan Biotech, Xiamen, China) according to the manufacturer's instructions. Briefly, the swab-containing collection tubes were equilibrated to room temperature. After a brief vortexing and a simple spin, 1.0 mL of the swab eluent was transferred to a 1.5-mL microcentrifuge tube and spun at 20,000 × $g$ for 10 min. The supernatants were discarded and the pellets were resuspended in 1.0 mL of phosphate-buffered saline (PBS) and then loaded onto the instrument for automatic DNA extraction.

**Statistical analysis.** Categorical variables were expressed as numbers with percentages, and continuous variables were expressed as medians with interquartile ranges (IQR). The chi-squared ($\chi^2$) statistic or Fisher's exact test was used to determine $P$ values for categorical variables. The NARs of microorganisms were compared using the Mann-Whitney $U$ test. Results with two-sided $P$ values less than 0.05 were considered significant. Analyses were conducted using SPSS Statistics version 23.0 (IBM Corporation, Armonk, NY, USA) and open-source R software version 3.6.0 (R Foundation for Statistical Computing, Vienna, Austria).

**Logistic regression analysis.** The logistic regression models were constructed using the qualitative data, quantitative data, and the combination of both from the derivation cohort. In the qualitative data, independent variables were binary, where "1" indicates positive and "0" indicates negative. In the quantitative data, the microbial NAR that maximally discriminates case-control status was used to derive the cutoff value through receiver operating characteristic (ROC) analysis. The optimal cutoff values were identified at the maximum of Youden's $J$ statistic (38). Independent variables were binary, where "1" indicates a NAR above the cutoff value and "0" indicates a NAR below the cutoff value. Regarding the combination of qualitative and quantitative data, the candidate independent variables included all those derived from the qualitative and quantitative data. Univariable analysis was used to generate crude odds ratios (ORs) with 95% confidence intervals (CIs). The variables with a significance level at a $P$ value of <0.10 were included in the multivariable model. During the multivariable analysis, a stepwise selection method was used to identify potential variables with $P$ values of <0.05. Adjustment was made by age categories as a confounding variable (5-year age groups), and the adjusted odds ratios (aORs) are presented with 95% CIs. The model performance was assessed by calculating the areas under the curves (AUCs) using ROC analysis (MedCalc version 19.0.4; MedCalc Software Ltd., Ostend, Belgium). For multiple comparisons, the AUCs of the prediction models derived from the qualitative data, quantitative data, and their combination were compared using the DeLong test (39) with adjustment of $P$ values by the Benjamini-Hochberg method (40). The final prediction models derived from the combined qualitative and quantitative data were assessed by the validation cohort. The AUCs generated from the derivation and validation cohorts were compared and the calibration of the models was carried out using the Hosmer-Lemeshow goodness-of-fit test, with $P$ values greater than 0.05 indicating an acceptable fit.

**Random forest classification.** The random forest classifier was employed to determine the importance of STI pathogens in cervicitis and vaginitis. Both qualitative and quantitative data from all the microorganisms and age groups were included as candidate features to build this classifier using the R package random forest (41). Prior to model development, the derivation group (944 samples) was randomly split into training and test data sets, representing 70% and 30% of the total samples, respectively. Preliminary models were constructed with all features. Given that the prediction models may be less accurate when all features are included, the features were then selected using the R package Boruta. Following the feature selection, the models were verified by the test data set, and the evaluated feature importance was ranked according to the mean decrease Gini score using the varSelRF package. Finally, the models were assessed by the validation cohort regarding the diagnostic performance in terms of accuracy, sensitivity, specificity, and AUC using the R package pROC.

**Ethics approval.** The collection of specimens and associated clinical data used in this study was approved by the Research Ethics Committee of Xiamen University (no. XDYX2021016). All of the experiments were conducted in compliance with the Declaration of Helsinki, and informed consent was obtained from the individual patients or their guardians. Patients' data confidentiality was fully respected during data collection.

**Data availability.** Data are available upon request. The R source code can be found in the supplemental material.

## SUPPLEMENTAL MATERIAL

Supplemental material is available online only.
**SUPPLEMENTAL FILE 1**, PDF file, 0.8 MB.

## ACKNOWLEDGMENTS

We thank the clinical and laboratory teams of the Liuzhou Maternal and Child Health Hospital, First Affiliated Hospital of Xiamen University, and Xiamen Maternal and Child Health Hospital for their assistance. We gratefully acknowledge Xiaomei Huang for her excellent laboratory work on HPV testing. We thank Xilin Zhao for critical reading of the manuscript.

We have no conflict of interest to declare.

The funders had no role in study design, data collection, data analysis, interpretation, or writing of the manuscript. This work was supported by the Science and Technology Project of Xiamen (3502Z20183007 to Q.L.), Fujian Provincial Science and Technology Innovation Joint Fund Project (2021Y4001 to Q.L.), Key Bio-safety Research and Development Technology Project (2021YFC1200200 to Q.L.), and Fujian Provincial Major Science and Technology Project (2019Y4002 to Y.L.).

W.-T.K., Y.L., Y.X., Q.H., and Q.L. designed the study. W.-T.K., H.X., and Q.G. collected the data. W.-T.K. and Y.L. performed the statistical analyses and interpretations. Q.L. supervised the study. W.-T.K. wrote the first draft of the manuscript. All authors contributed to the critical revision of the manuscript.

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
