## [Reviewer comments · Microbiology Spectrum]

Microbiology Spectrum

Qualitative and quantitative detection of multiple sexually transmitted infection pathogens reveals distinct associations with cervicitis and vaginitis

Wen-Tyng Kang, Haibo Xu, Yiqun Liao, Qiwei Guo, Ye Xu, Qiuying Huang, and Qingge Li

Corresponding Author(s): Qingge Li, Xiamen University

Review Timeline:

Submission Date:	June 2, 2022
Editorial Decision:	August 11, 2022
Revision Received:	September 9, 2022
Accepted:	October 5, 2022

Editor: Meghan Starolis

Reviewer(s): The reviewers have opted to remain anonymous.

Transaction Report:

DOI: <https://doi.org/10.1128/spectrum.01966-22>

August 11, 2022

Prof. Qingge Li
Xiamen University
School of Life Sciences
Huangchaoyang Buliding E202
Xiang'an Campus Xiamen University
Xiamen, Fujian 361102
China

Re: Spectrum01966-22 (Qualitative and quantitative detection of multiple sexually transmitted infection pathogens reveals distinct associations with cervicitis and vaginitis)

Dear Prof. Qingge Li:

Thank you for submitting your manuscript to Microbiology Spectrum. The review process is now complete, and the reviewers have some major concerns as well as minor feedback which must be addressed before being considered for publication. When submitting the revised version of your paper, please provide (1) point-by-point responses to the issues raised by the reviewers as file type "Response to Reviewers," not in your cover letter, and (2) a PDF file that indicates the changes from the original submission (by highlighting or underlining the changes) as file type "Marked Up Manuscript - For Review Only". Please use this link to submit your revised manuscript - we strongly recommend that you submit your paper within the next 60 days or reach out to me. Detailed instructions on submitting your revised paper are below.

Link Not Available

Sincerely,

Meghan Starolis

Journals Department
Reviewer comments:

Reviewer #1 (Comments for the Author):

The authors tested a cohort of >1500 women with a commercially available 16-plex MeltArray and laboratory-developed quantitative RT-PCR assays for STIs. Each participant was also assessed for cervicitis and vaginitis

Overall, the study used appropriate statistical tests to develop associations between different STIs, cervicitis, and vaginitis. I

believe the author's make some interesting associations here, and of interest is the layering of qualitative and quantitative information on STIs for developing predictive models. One of the limitations of this study (acknowledged by the authors) is the absence of information on behavioral/lifestyle factors that affect incidence of cervicitis and vaginitis, and the exclusion of microbiome markers. Inclusion of these factors into their models may have further improved the predictive value of their models.

The authors state, "STI pathogen identification and quantification can be used to diagnose cervicitis and also help improve correct diagnosis of vaginitis"; however, it was unclear from this work how diagnosis could be improved vs. traditional methods or emerging alternatives such as microbiome analyses. Perhaps this could be elaborated a bit further in the manuscript, highlighting diagnostic gaps that could be addressed by this assay.

I appreciate the author's inclusion of additional data in the study, including the original R code and complete list of qPCR primers and their sources. These should provide valuable resources for other research teams should they attempt to replicate aspects of this study.

Some comments about calculation of LOD:

The authors state: "To determine the limit of detection (LOD) for each target, 10-fold serially diluted plasmid DNA templates, ranging from 1.0×10^7 copies/ μ L to 1.0×10^0 copies/ μ L, were detected in **triplicate** for each concentration. The lowest concentration level with a detection rate of 95% from **20 replicate** detection was defined as the LOD."

It is unclear from above if each step of the dilution series was performed in triplicate, or 20 replicates. Please clarify.

The authors define their LOD as quantitative range as: "The lowest concentration level with a detection rate of 95% from 20 replicate detection was defined as the LOD. The quantification range was defined as one order of magnitude above the LOD." However, all replicates displayed a quantitative range above 10^1 copies, meaning (according to the authors' definition) all had an LOD, or 95% detection, at 10^0 copies. As this was the final step of their dilution series, it appears all dilutions tested were above the LOD. Because of this, the authors cannot really make any statements regarding the limit of detection. Rather, to comment on the true LOD they should continue dilutions until no detection is recorded (likely 10⁻¹) and perform a probit analysis. With that said, for the purpose of this study simply showing the quantitative range of the assay should suffice.

Some minor comments:

Line 61: Please elaborate on "these STIs have been on a steady rise recently". This statement is vague and should include specifics on how much of an increase over what period.

Line 68: Please provide reference for "syndromic management in the absence" and provide a specific number for the alarming rate.

Line 74: Please rephrase "many STIs show no symptom" to "many STIs present asymptotically".

Line 91: should read "predictive models" not "predicative models".

Line 134: *U. parvum* has been shown to be a normal constituent of the microbiome in healthy women of reproductive age; however, here it is defined solely as a 'pathogen'. This would explain why it was present in so many of the healthy controls and was only predictive when measured quantitatively. I may clarify this in the results/discussion.

Reviewer #2 (Comments for the Author):

Summary:

The authors aim to define the relationships between the presence and abundance of different pathogens and their association with cervicitis and vaginitis. Some of these are of known significance with disease pathogenesis. This is a very interesting article, well structured, showing interesting and novel associations with different pathogens including sexually transmitted infections that could aid in the diagnosis of cervicitis and vaginitis.

Major issues:

- Please state the rationale to include CMV, HHV8 and VZV on a STI screening assay. Even though CMV and HHV8 agents are transmitted through different body fluids including semen, they are not considered to be STIs. VZV rarely infects genital areas and the main mode of transmission is direct contact with vesicle fluid and inhalation of airborne particles. These could have been found in this study as bystanders and could be shedding since all these are herpesviruses that will become latent and can periodically reactivate. Since PCR is not able to differentiate between primary infection and reactivation, I would suggest separating those agents from the true/proved STI causing pathogens.

- It would be important to add in the rationale for this study, how the authors would propose this test could be used clinically and what the clinical significance of these findings would be for the treatment and better patient management.

Minor issues:

- Line 32 typo on "patients"

- Line 368 - "extracted" instead of "abstracted"
- Fig4/5 sensitivity not "sensitivity"

Staff Comments:

Preparing Revision Guidelines

Please return the manuscript within 60 days; if you cannot complete the modification within this time period, please contact me. If you do not wish to modify the manuscript and prefer to submit it to another journal, please notify me of your decision immediately so that the manuscript may be formally withdrawn from consideration by Microbiology Spectrum.

Review manuscript: Qualitative and quantitative detection of multiple sexually transmitted infection pathogens reveals distinct associations with cervicitis and vaginitis

Authors: Kang WT, et al.

Summary:

The authors aim to define the relationships between the presence and abundance of different pathogens and their association with cervicitis and vaginitis. Some of these are of known significance with disease pathogenesis. This is a very interesting article, well structured, showing interesting and novel associations with different pathogens including sexually transmitted infections that could aid in the diagnosis of cervicitis and vaginitis.

Major issues:

- Please state the rationale to include CMV, HHV8 and VZV on a STI screening assay. Even though CMV and HHV8 agents are transmitted through different body fluids including semen, they are not considered to be STIs. VZV rarely infects genital areas and the main mode of transmission is direct contact with vesicle fluid and inhalation of airborne particles. These could have been found in this study as bystanders and could be shedding since all these are herpesviruses that will become latent and can periodically reactivate. Since PCR is not able to differentiate between primary infection and reactivation, I would suggest separating those agents from the true/proved STI causing pathogens.
- It would be important to add in the rationale for this study, how the authors would propose this test could be used clinically and what the clinical significance of these findings would be for the treatment and better patient management.

Minor issues:

- Line 32 typo on "patients"
- Line 368 – "extracted" instead of "abstracted"
- Fig4/5 sensitivity not "sensitivity"

Point-to-Point Response to Reviewers

Reviewer #1 (Comments for the Author):

The authors tested a cohort of >1500 women with a commercially available 16-plex MeltArray and laboratory-developed quantitative RT-PCR assays for STIs. Each participant was also assessed for cervicitis and vaginitis

Overall, the study used appropriate statistical tests to develop associations between different STIs, cervicitis, and vaginitis. I believe the author's make some interesting associations here, and of interest is the layering of qualitative and quantitative information on STIs for developing predictive models. One of the limitations of this study (acknowledged by the authors) is the absence of information on behavioral/lifestyle factors that affect incidence of cervicitis and vaginitis, and the exclusion of microbiome markers. Inclusion of these factors into their models may have further improved the predictive value of their models.

The authors state, "STI pathogen identification and quantification can be used to diagnose cervicitis and also help improve correct diagnosis of vaginitis"; however, it was unclear from this work how diagnosis could be improved vs. traditional methods or emerging alternatives such as microbiome analyses. Perhaps this could be elaborated a bit further in the manuscript, highlighting diagnostic gaps that could be addressed by this assay.

Response: We agree with our reviewer's constructive suggestion. We have now elaborated this point in the "Discussion" to highlight diagnostic gaps that could be addressed by this assay (lines 321–328).

I appreciate the author's inclusion of additional data in the study, including the original R code and complete list of qPCR primers and their sources. These should provide valuable resources for other research teams should they attempt to replicate aspects of this study.

Response: Thanks for pointing this out!

Some comments about calculation of LOD:

The author's state: "To determine the limit of detection (LOD) for each target, 10-fold serially diluted plasmid DNA templates, ranging from 1.0×10^7 copies/ μ L to 1.0×10^0 copies/ μ L, were detected in triplicate for each concentration. The lowest concentration level with a detection rate of 95% from 20 replicate detection was defined as the LOD."

It is unclear from above if each step of the dilution series was performed in

triplicate, or 20 replicates. Please clarify.

Response: In order to determine the LOD, each serial dilution was initially detected in triplicate to find an approximate LOD. Confirmation of LOD was further achieved by testing 20 replicates of a target at their approximate LOD concentration. The final LOD was concluded when at least 19 of the 20 replicates gave positive results ($19/20 \times 100 = 95\%$). We clarify this point by revising the original description of the above procedure (lines 403–409).

The authors define their LOD as quantitative range as: "The lowest concentration level with a detection rate of 95% from 20 replicate detection was defined as the LOD. The quantification range was defined as one order of magnitude above the LOD."

However, all replicates displayed a quantitative range above 10^1 copies, meaning (according to the authors' definition) all had an LOD, or 95% detection, at 10^0 copies. As this was the final step of their dilution series, it appears all dilutions tested were above the LOD. Because of this, the authors cannot really make any statements regarding the limit of detection. Rather, to comment on the true LOD they should continue dilutions until no detection is recorded (likely 10^{-1}) and perform a probit analysis. With that said, for the purpose of this study simply showing the quantitative range of the assay should suffice.

Response: We agree with our reviewer's comments that we cannot solidly define LOD as the lowest concentration we tested still generated 95% positive signal. Since theoretically an LOD cannot be lower than 1 copy/assay, the lowest concentration we used in our LOD determination was 10^0 copy/uL. Had we included a 10^{-1} copy/uL dilution in our LOD measurement, the LOD, most of which happened to be the lowest concentration we tested (e.g., 10^0 copy/uL), would have been firmly established experimentally. Following our reviewer's suggestion, we have revised the statement for the quantitative range of target quantification (lines 409–412).

Some minor comments:

Line 61: Please elaborate on "these STIs have been on a steady rise recently". This statement is vague and should include specifics on how much of an increase over what period.

Response: The statement has been amended to include "These STIs have been on a steady rise to 689 million in 2019" (line 60–61) with an added reference (ref 3).

Line 68: Please provide reference for "syndromic management in the absence" and provide a specific number for the alarming rate.

Response: We now add the references for "syndromic management in the absence" (refs 7, 8, and 9). We also provide the misdiagnosis rate for STIs in the revised manuscript (lines 68–70).

Line 74: Please rephrase "many STIs show no symptom" to "many STIs present asymptotically".

Response: Done as suggested (lines 75–76).

Line 91: should read "predictive models" not "predicative models".

Response: Done as suggested (line 92).

Line 134: *U. parvum* has been shown to be a normal constituent of the microbiome in healthy women of reproductive age; however, here is it defined solely as a 'pathogen'. This would explain why it was present in so many of the healthy controls and was only predictive when measured quantitatively. I may clarify this in the results/discussion.

Response: We have added the suggested description in the "Discussion" (lines 280–282).

Reviewer #2 (Comments for the Author):

Summary:

The authors aim to define the relationships between the presence and abundance of different pathogens and their association with cervicitis and vaginitis. Some of these are of known significance with disease pathogenesis. This is a very interesting article, well structured, showing interesting and novel associations with different pathogens including sexually transmitted infections that could aid in the diagnosis of cervicitis and vaginitis.

Major issues:

- Please state the rationale to include CMV, HHV8 and VZV on a STI screening assay. Even though CMV and HHV8 agents are transmitted through different body fluids including semen, they are not considered to be STIs. VZV rarely infects genital areas and the main mode of transmission is direct contact with vesicle fluid and inhalation of airborne particles. These could have been found in this study as bystanders and could be shedding since all these are herpesviruses that will become latent and can periodically reactivate. Since PCR is not able to differentiate between primary infection and reactivation, I would suggest separating those agents from the true/proved STI causing pathogens.

Response:

The purpose of this study is to investigate the roles of a wide selection of relevant STI pathogens in cervicitis and vaginitis. The “pathogen lists” included in our study is coming from the commercial MeltArray STIs assay, the target number of which is among the largest in the market. To address this concern regarding the rationale to include CMV, HHV8, and VZV on a STI screening assay, we consulted the manufacture company Zeesan Biotech. We were told that their pathogen inclusion rule is: 1) WHO guidelines on STIs, 2) published guidelines and expert consensus on STIs; 3) emerging pathogens recently reported for STIs.

We checked them up with the website of WHO and noticed that there is a statement that “there are over 30 different sexually transmissible bacteria, viruses, and parasites” under the topic of STIs (<https://www.who.int/europe/news-room/questions-and-answers/item/sexually-transmitted-infections>), where CMV is among the list. A full list of these STI pathogens can be found in Table 1.1 in a book entitled *ABC of Sexually Transmitted Infections*, 6th edition. Edited by Karen E. Rogstad, 2011 Blackwell Publishing Ltd. p2, where both CMV and HHV-8 are included under the category of “Viral infections”.

As VZV is not listed in the above references, we further searched VZV in

PubMed and noticed the following articles: 1) Birch et al reported that VZV was present in nearly 3% of patients presenting with genital infections. The authors proposed that VZV should be considered as part of the differential diagnosis in patients who are evaluated for HSV genital infection; 2) Granato et al reported detection of 11.1% of positive VZV from male and female genital sites, suggesting that the appearance of zoster lesions in the genital area may not be as uncommon as previously thought and that this finding would have considerable impact on patient counseling and public health considerations.

Based on reviewer's concern and the above searching results, we have added the following sentences in the "Discussion":

"Third, the pathogens included in our study are all from the MeltArray STIs assay. Although many of them are proved, classical STI-causing pathogens, some (e.g., VZV and HHV8), although had been implicated in some STIs, remained to be confirmed. Their exact roles in STI needs further study." (lines 351–354):

References cited:

1. Birch CJ, Druce JD, Catton MC, MacGregor L, Read T. 2003. Detection of varicella zoster virus in genital specimens using a multiplex polymerase chain reaction. *Sex Transm Infect* 79:298–300.
2. Granato PA, DeGilio MA, Wilson EM. 2016. The unexpected detection of varicella-zoster virus in genital specimens using the Lyra™ Direct HSV 1+2/VZV Assay. *J Clin Virol* 84:87–89.

- It would be important to add in the rationale for this study, how the authors would propose this test could be used clinically and what the clinical significance of these finding would be for the treatment and better patient management.

Response: Understanding the association between the studied pathogens and the two diseases could help develop pathogen-oriented molecular testing to aid and improve current syndrome-based diagnosis. This is particularly true for cervicitis, which showed a sensitivity of 92% and a specificity of 98% in the validation sample of the random forest model. As for vaginitis, we have added the sentences describing the rationale for the clinical use of our study (lines 321–328).

Regarding the clinical significance of our finding would be for the treatment and better management, we add the following sentences in the "Discussion":

"Rapidly identifying the exact disease-causing pathogen(s) would enable

appropriate drug selection, prognosis predication, progress monitoring, and even prevention measures planning, thus leading to greatly improved patient management for STIs.” (lines 343–346)

Minor issues:

Line 32 typo on "patients"

Response: Corrected as suggested (line 32).

Line 368- "extracted" instead of "abstracted"

Response: Revised as suggested (line 379).

Fig4/5 sensitivity not "sencitivity"

Response: Corrected as suggested (Fig 4/5).

October 5, 2022

Prof. Qingge Li
Xiamen University
School of Life Sciences
Huangchaoyang Buliding E202
Xiang'an Campus Xiamen University
Xiamen, Fujian 361102
China

Re: Spectrum01966-22R1 (Qualitative and quantitative detection of multiple sexually transmitted infection pathogens reveals distinct associations with cervicitis and vaginitis)

Dear Prof. Qingge Li:

Your manuscript has been accepted, and I am forwarding it to the ASM Journals Department for publication. You will be notified when your proofs are ready to be viewed.

Sincerely,

Meghan Starolis
Editor, Microbiology Spectrum

Journals Department
Supplemental material: Accept